# Performing a Sonar Acceptance Test of the Kongsberg EM712 Using Open-Source Software: A Case Study of Kluster

**Eric Younkin * and S. Harper Umfress ***

NOAA National Ocean Service, Office of Coast Survey, Hydrographic Systems and Technology Branch, Silver Spring, MD 20910, USA
* Correspondence: eric.g.younkin@noaa.gov (E.Y.); samuel.umfress@noaa.gov (S.H.U.)

**Abstract:** In the world of seafloor mapping, the ability to explore and experiment with a dataset in its raw and processed forms is critical. Kluster is an open-source multibeam data processing software package written in Python that enables this exploration. Kluster provides a suite of multibeam processing features, including analysis, visualization, gridding, and data cleaning. We demonstrated these features using a recently acquired dataset from a Kongsberg EM712 multibeam echosounder aboard NOAA Ship *Fairweather*. This test dataset served to illustrate the fundamental analysis abilities of the software, as well as its utility as a troubleshooting tool both in the field and during post-processing. Kluster has the capability to perform the Sonar Acceptance Test in full, including common experiments like the patch test, extinction test, and accuracy test, which are generally performed on new systems. When questions arise regarding the integration or parameter settings of a system, this software allows the user to quickly and clearly visualize much of the raw data and its associated metadata, which is a vital step in any investigative effort. With its emphasis on accessibility and ease of use, Kluster is an excellent tool for users who are inexperienced with multibeam sonar data processing.

**Keywords:** hydrography; multibeam; open-source; python; Kluster; hydrographic software



## 1. Introduction

The wealth of resources available for scientific processing and analysis in Python is growing every day. This includes the extensive library of algorithms found in SciPy [1], powerful n-dimensional data structures in NumPy [2], and, more recently, the Pangeo ecosystem [3], which includes packages such as Dask, Xarray, and Zarr. All these packages allow for rapid prototyping of applications and detailed analysis without the required effort to implement existing algorithms and data structures. If processed multibeam data were available in Python, these packages could be used by the scientific community to access the data in a way that is not currently available.

Kluster [4–6] is designed to thrive in this space. It relies on Zarr and Numpy for n-dimensional data structures in memory and on disk. It uses Xarray and Dask to support multiprocessing across all of its processing algorithms. With the core structure being the Xarray Dataset, scientists can read and operate on processed Kluster datasets without using Kluster, relying solely on the Xarray package. Using Zarr, the Kluster datasets are pre-chunked for efficient access over the internet, making the Kluster format an efficient archival format.

Several multibeam processing packages have already been developed and implemented in the open-source space. Most notably, MB-System [7], originally developed at the Lamont-Doherty Earth Observatory of Columbia University using the C language, and SonarScope [8], developed at Ifremer. MB-System is widely considered to be the best alternative to commercial software but has a steep learning curve and requires an understanding of scripting in Linux and command line usage [9]. SonarScope is developed in MATLAB,

and is available on Ifremer's GitLab repository, and provides a more complete graphically driven experience. While C and MATLAB are well established languages for development, Python is generally more accessible and an easier language to work in within the scientific community, where many individuals might not have a computer science degree.

To demonstrate the features and effectiveness of Kluster, this paper will outline the Sonar Acceptance Test (SAT) for the newly purchased Kongsberg EM712 multibeam echosounder (MBES) conducted on the NOAA Ship *Fairweather* in May and June of 2022. The SAT encompasses all integration, data acquisition, and processing that is required to ensure that the new sonar meets charting specifications [10]. This is generally a manual process, completed with a combination of Python scripts, commercial software, and minor software development when required. With Kluster, all SAT tests are integrated into the graphical interface, making them simple to run and visualize. The SAT provides a comprehensive test for a multibeam data processing software package, as there are often data issues that interfere with processing, and many custom needs for analysis outside of the standard workflow. As is outlined below, the unique capabilities of Kluster were leveraged throughout this project and proved vital in the qualification of the sonar system under evaluation.

## 2. Materials and Methods

All software developed in the Kluster project is available on GitHub for download and user contribution. Kluster relies on several custom submodules, that are also available on GitHub in separate repositories, shown in Table 1. Kluster provides build instructions in the documentation, as well as Windows builds for each new release.

**Table 1.** Kluster and submodule URLs.

| Module | URL |
| --- | --- |
| Kluster | https://github.com/noaa-ocs-hydrography/kluster (accessed on 22 September 2022) |
| bathycube | https://github.com/noaa-ocs-hydrography/bathycube (accessed on 22 September 2022) |
| vyperdatum | https://github.com/noaa-ocs-hydrography/vyperdatum (accessed on 22 September 2022) |
| drivers | https://github.com/noaa-ocs-hydrography/drivers (accessed on 22 September 2022) |
| bathygrid | https://github.com/noaa-ocs-hydrography/bathygrid (accessed on 22 September 2022) |

Kluster currently supports the Kongsberg .all and .kmall formats, with additional limited support for EK60 and EK80 systems, including a custom amplitude detection capability. Kluster also supports the Reson .s7k format, as is detailed in the Kluster documentation [5] section on 'Requirements'.

The raw dataset for the EM712 SAT is not currently available online, due to limitations with hosting large datasets that are not a part of the normal production chain.

### 2.1. Kluster—Theory of Operation

Kluster first relies on a conversion step, to pull records from the raw multibeam format to an intermediate custom format that was designed for Kluster. This format is stored on disk as Zarr arrays and can be loaded in Xarray as a Dataset, sorted by time and beam. Conversion will automatically sort incoming data into containers, where each container is a specific sonar model, date and sonar serial number. As an example, the extinction test for this experiment exists within container "em712_10070_05_10_2022", which is the container for the EM712 with serial number 10070 on 10 May 2022. Having this organization allows the user to drag-and-drop files into Kluster without concern for which day or sonar they originate from, information that is oftentimes not clear to the end user that was not involved during acquisition.

The initial stages of processing in Kluster are heavily inspired by existing academic research on post-processing multi-sector multibeam systems [11]. Using vessel attitude and mounting angle offsets for the sonar, Kluster corrects the original array-relative beam angles and saves the corrected angle and azimuth to disk, as illustrated in Figure 1. This

process currently assumes the transmit and receive arrays are concentric, which may create issues in deeper water, and is a current area of academic interest [12,13]. These values are used during sound velocity correction to calculate the correct offsets from reference point to beam end point, which is then used during georeferencing to build the three-dimensional point cloud. The products of these processing steps are then used in Kluster's Total Propagated Uncertainty (TPU) model, which was built following guidance from the paper on the multibeam uncertainty model [14].

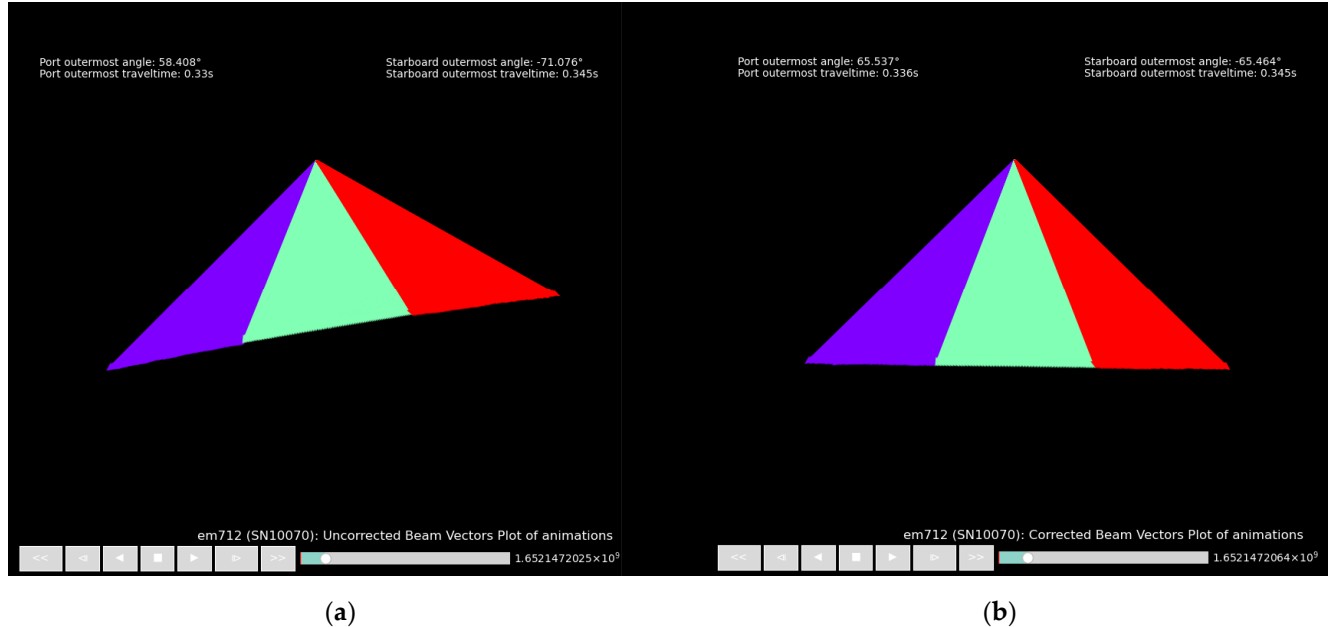

**Figure 1.** Kluster animations of uncorrected and corrected beam vectors available in the Basic Plots tool. Illustrates: (**a**) Raw beam angles as seen in the multibeam data format and; (**b**) Raw angles corrected for attitude and mounting angles as a result of Kluster processing. Colored by multibeam sector.

The user may elect to visualize or manually remove any outliers using the Points View, which displays the point cloud in two or three dimensions. Kluster also provides a filtering utility with some custom filters provided, as well as support for custom filter plugins that can be created by the user for their specific needs.

With a processed point cloud, the user can generate grids using the bathygrid module, which builds single or variable resolution tiles, again saved to disk using Zarr and Xarray. These grids support larger-than-memory datasets, store both points and cell values, support updates through adding and removing additional datasets, and allow for exporting to common GDAL formats.

### 2.2. NOAA Ship Fairweather & the Kongsberg EM712—Background

NOAA Ship *Fairweather* is 231-foot long hydrographic survey vessel homeported in Ketchikan, Alaska (Figure 2). Commissioned in 1968, the ship operates an EM712 sonar and carries a variety of small boats with individual sonars and additional charting capabilities. The Kongsberg EM712 installed on NOAA Ship *Fairweather* is a 0.5° × 1.0° system with a specified maximum depth of 3200 m and a maximum coverage of 3950 m. This system is controlled by the latest version of Kongsberg's SIS5 software and is one of the first instances of this software in the NOAA fleet. There were several integration issues with this software that were resolved prior to sailing, mostly centered around interfacing with other software packages. The EM712 receives attitude, velocity, and navigation from the Applanix POS MV installed on the vessel. The EM712 transmitter serves as the vessel reference point, and all offsets and angles are relative to it. These offsets and angles are entered into SIS

and the POS MV setup screens such that the raw multibeam data is logged with all the correct values.

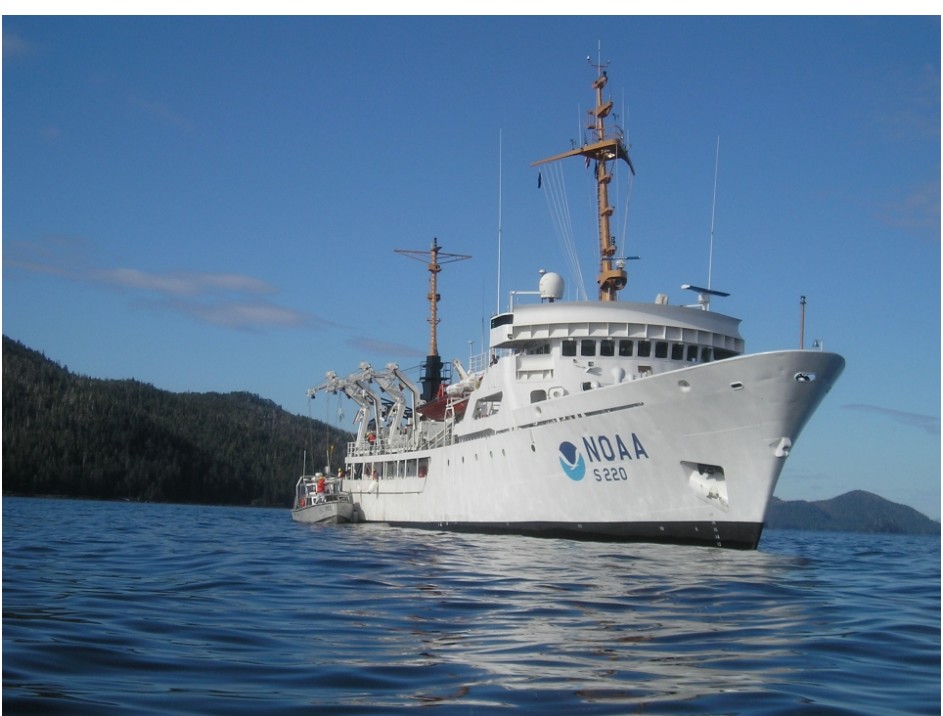

**Figure 2.** Image of NOAA Ship *Fairweather*, courtesy of NOAA.

*2.3. Ancillary Data Processing*

During this project, sound velocity profiles were acquired using a Moving Vessel Profiler (MVP) winch system and an AML Micro CTD sensor. These raw profiles are processed in Sound Speed Manager, which is an open-source sound velocity processing software available through Hydroffice [15]. Kluster supports importing these processed sound velocity text files as additional profiles to those currently in the raw multibeam data. Sound velocity profiles are used during sound velocity correction in Kluster based on one of the available selection algorithms seen in the Kluster project settings.

Raw POS MV data is processed in Applanix POSPac using the Trimble RTX corrector service to produce processed GNSS/INS data in the Applanix Smoothed Best Estimate of Trajectory (SBET) format. Kluster can import the processed navigation and ellipsoid height for use in all georeferencing operations, which is of particular significance with ellipsoidally referenced surveying (ERS) techniques where the final depth is a product of the ellipsoid height [16]. Processed navigation is generally used throughout all vertical datum selections in Kluster.

## 3. Results

The EM712 SAT took place off the coast of San Francisco, California, USA. All data were converted in Kluster by simply dragging in the raw Kongsberg KMALL files and using the start button in the Action pane to commence conversion. These files are shown in the screenshot below in Figure 3 with the project information on the bar on the left, and the tracklines shown in the embedded QGIS map view on top of an OpenStreetMap WMS layer. By utilizing QGIS tools for the map view in Kluster, resources such as web map services and generic raster and vector format support are made available in a simple and intuitive way.

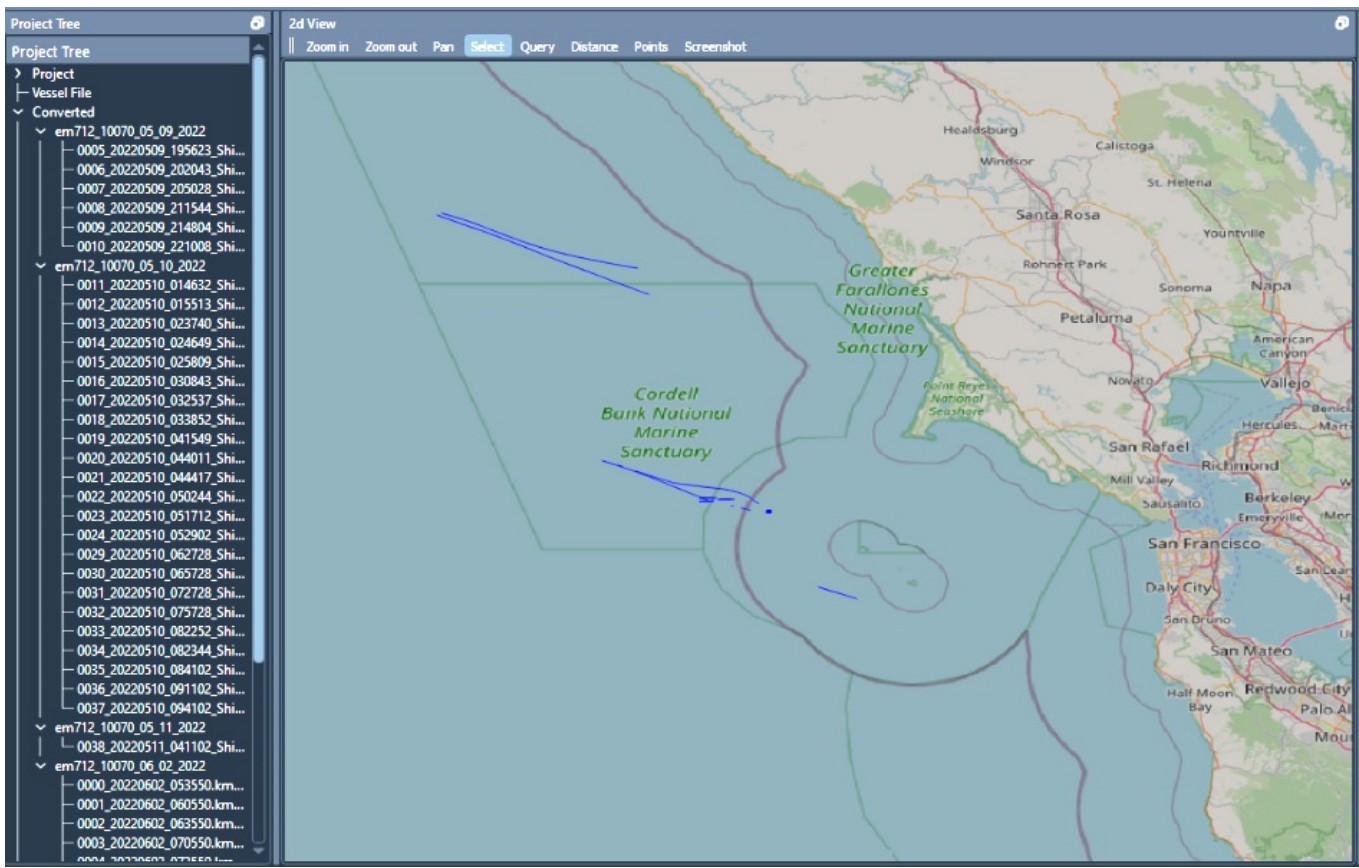

**Figure 3.** Map view of the project area in Kluster, multibeam tracks shown in blue.

Kluster has a custom state, machine-driven processing system called the Intelligence module, which generates processing actions based on the state of the data and the desired processing settings. Dragging in new multibeam files that do not exist in the existing containers will generate a new conversion action. Including additional sound velocity profiles will generate a new import sound velocity action. If sound velocity processing had occurred once already, importing new profiles would generate a re-sound velocity correct action. The Intelligence module will ensure that the data is fully processed as project settings change and new data is added. For this dataset, we processed to the NOAA mean lower low water (MLLW) datum, using the processed ellipsoid height from the SBET and the vyperdatum module to automatically generate a separation model between ellipsoid and MLLW, as shown in Figure 4.

With the newly processed data, we are now able to proceed to the SAT tests. NOAA's SAT procedure generally includes the following tests, which will dictate the layout of the rest of this section:

- Offsets and Integration
- Patch Test (Boresight Angle Estimation)
- Extinction Test (Range Test)
- Accuracy Test (Vertical Accuracy Test)

### 3.1. Offsets and Integration

Kluster includes on conversion all offsets and supporting parameters that SIS can provide in the KMALL file. These are shown in the container attribution in the Attribute window but are primarily interacted with through the Vessel Setup utility. The Vessel Setup utility allows the user to see and change the offsets and setup parameters within the container selected. Additionally included is a few 3D models of ships that can be used as a reference for the blocks that represent the sensor locations. The user can also include a 3D

model for their vessel to visualize the sensor locations in OBJ format. Figure 5 shows the sonar transmitter and receiver for this survey, with the offsets from the vessel reference point on the left.

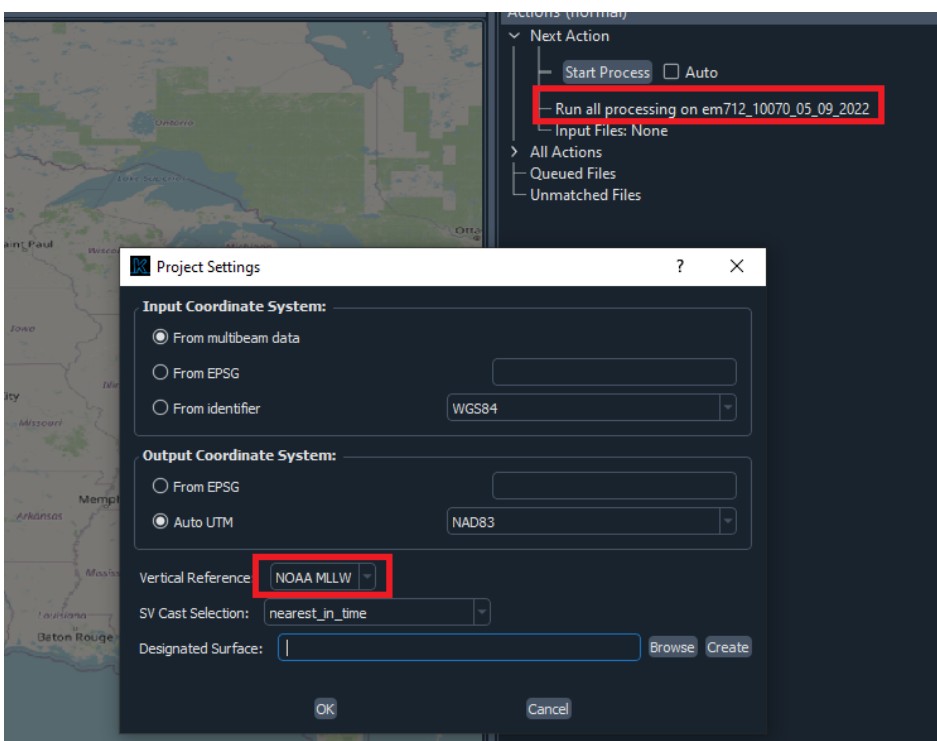

**Figure 4.** Project settings with desired vertical reference and the "Run all processing" action that is spawned as a result.

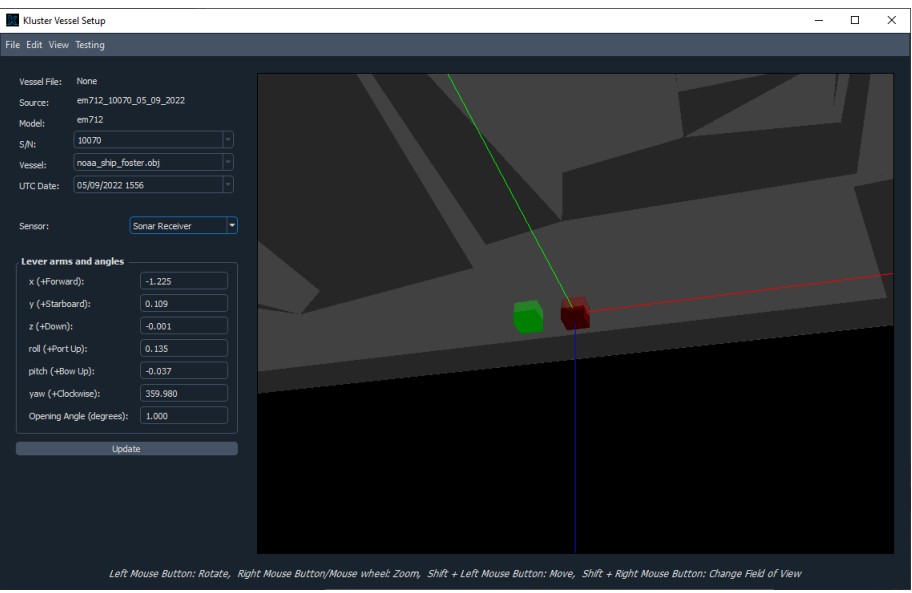

**Figure 5.** Vessel Setup utility showing the sonar transmitter (red) and receiver (green).

Changes to these parameters will spawn the appropriate processing action, depending on the value altered. Mounting angle changes require a full reprocessing of the dataset, while changes to uncertainty parameters will only spawn an uncertainty processing step. These values in the Vessel Setup match the transmitter-relative values entered into SIS, so no further action is required.

### *3.2. Patch Test*

The patch test, or boresight angle estimation, includes six survey lines run on 9 May 2022, that are processed and evaluated to determine any residual angular offsets or timing offset between the sonar transducer and the motion sensing unit. These latency, pitch, roll, and yaw offsets are determined by comparing the bathymetry of lines collected at different orientations over both flat and sloped seafloor. As a result, we initially needed to generate a processed grid to assess the acquired bathymetry. Figure 6 illustrates this, showing an 8.0 m resolution Combined Uncertainty and Bathymetry Estimator (CUBE) [17] grid, with data processed to the NOAA MLLW ERS datum. CUBE is provided in the bathycube module that was developed alongside Kluster. Depths range from 100 m to 700 m.

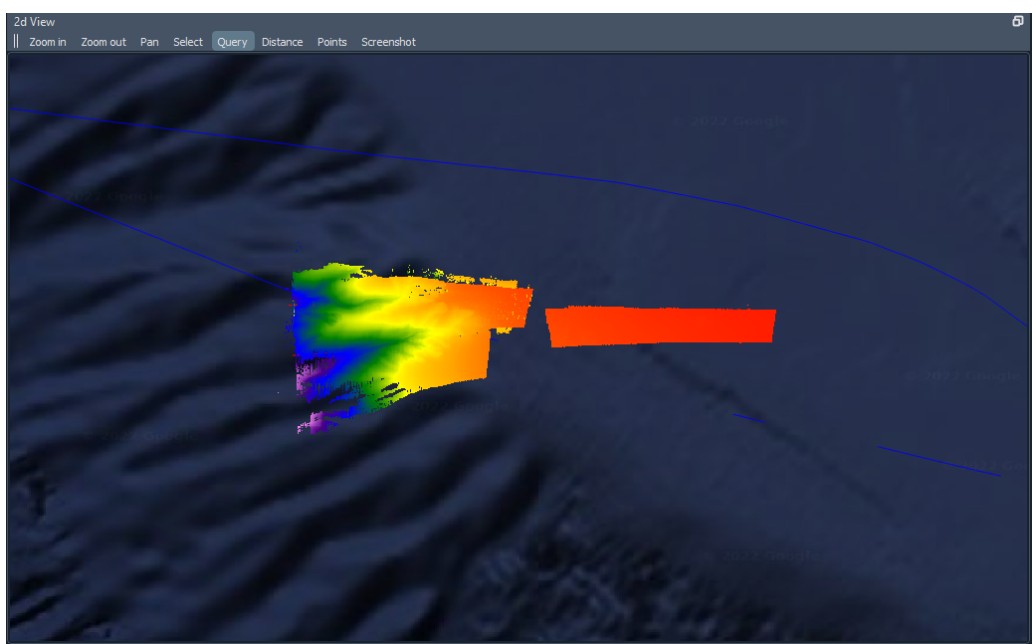

**Figure 6.** Patch Test Area with an 8.0 m resolution gridded dataset of the processed multibeam data.

The latency test involves isolating a line over flat seafloor that was acquired with, ideally, a significant amount of roll. This dataset can be analyzed using the Kluster Advanced Plots—Wobble Test tool to determine if there is a correlation between the roll rate and the ping slope, where the slope of the regression would be the calculated latency between sonar and motion sensor. In the case of line 0010, we were unable to determine any significant latency value. If we had computed a value of several milliseconds or greater, we would enter it into SIS, before commencing any other tests. Alternatively, the value can be added in Kluster when post-processing the dataset. Figure 7 illustrates the Latency Test as completed in Kluster.

The remaining three elements of the Patch Test can be determined using the Kluster Patch Test tool. Roll is calculated using the same line run twice in opposite directions over a flat seafloor, pitch is calculated using the same line run twice in opposite directions over a slope, and yaw involves two lines run in the same direction down a slope offset from each other. These lines can be chosen from the six included in the Patch Test dataset, which were specifically acquired to meet these guidelines. To accomplish the Patch Test, new values are chosen by the user and entered into the utility, to reprocess the data displayed in the point cloud viewer until the data is visually determined to be in alignment and acceptable. Figure 8 shows this process, with a narrow slice of the dataset perpendicular to the vessel motion shown in the points view, colored by the line of origin.

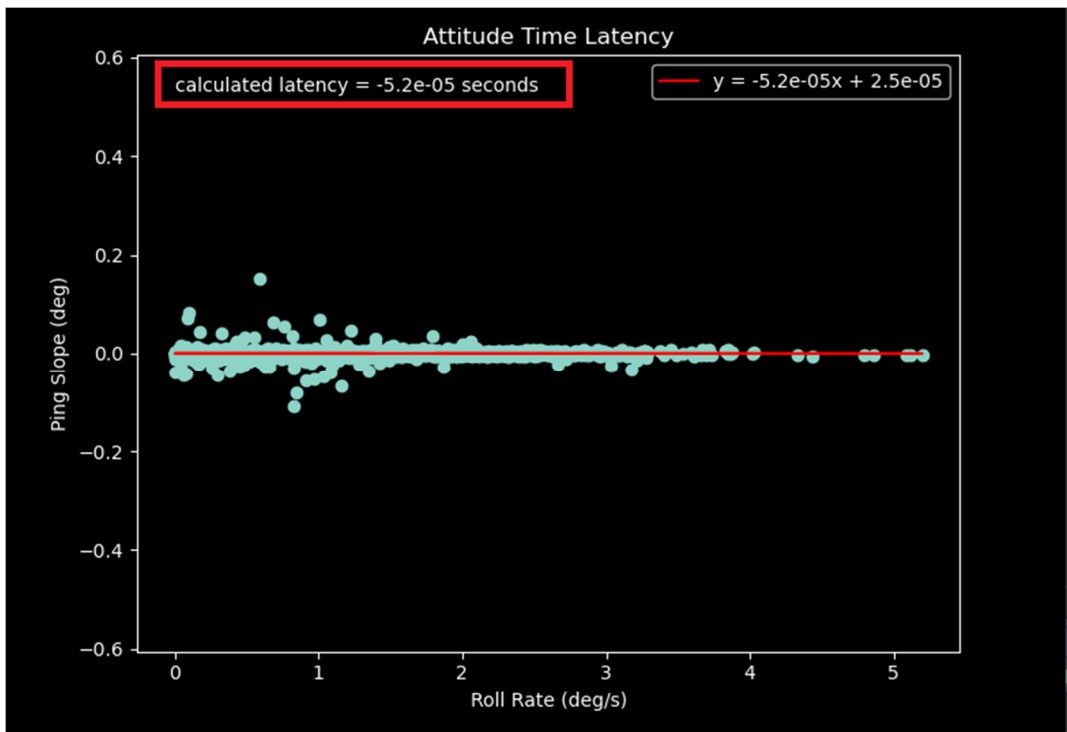

**Figure 7.** Latency Test shown in the Advanced Plots tool.

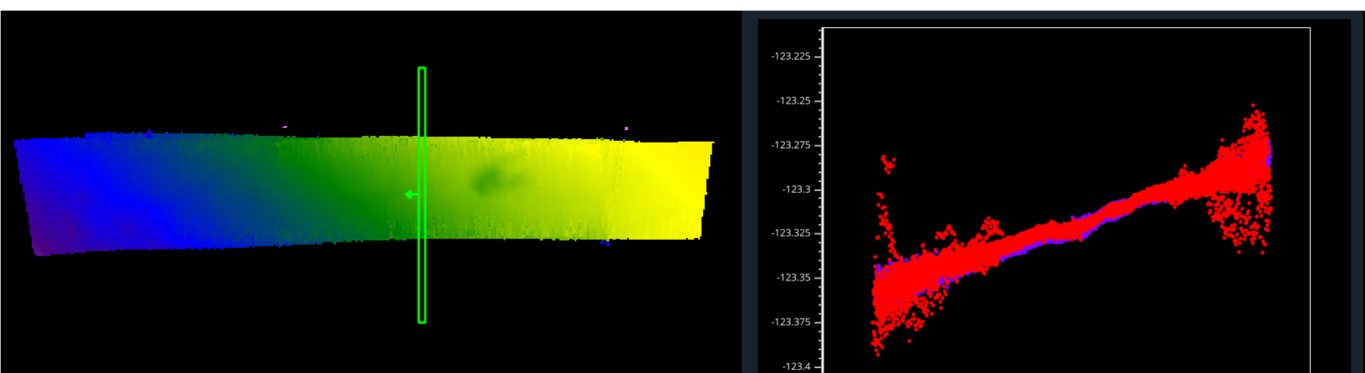

**Figure 8.** Patch Test utility shown, with data being assessed for roll mounting angle offset.

If new values are found, they can be entered into the Vessel Setup utility for reprocessing the existing dataset; though ultimately, they should be entered in SIS, such that the raw data will already have the correct mounting angles.

### 3.3. Extinction Test

The extinction test serves to determine the effective swath width of the system throughout the expected depth range, as well as the system's ability to automatically select the appropriate depth-dependent settings. The survey lines are generally run from shallow to deep and then deep to shallow following the reciprocal course. The resulting data can be plotted using the Kluster Advanced Plots tool to visualize the outermost beams seen for each depth range. The extinction test area and acquired bathymetry are shown in Figure 9 below. After the completion of one extinction test, we noted non-uniform changes in the depth settings and underperformance of the sonar. We used Kluster to diagnose the issue and subsequently reacquired the dataset.

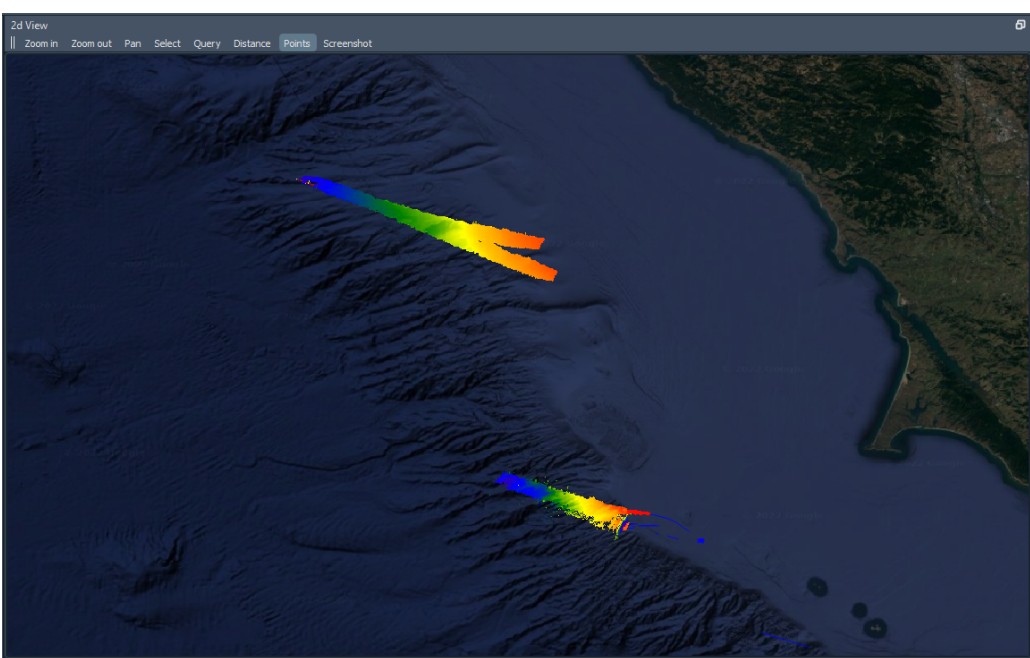

**Figure 9.** Extinction test area and gridded datasets with 64 m resolution shown on top of the Satellite WMS layer.

The extinction test results are a series of plots showing the width of the swath at different depths and colored by sonar mode setting or frequency. The first extinction test results are shown in Figure 10 below. Each of the three plots shows the progression from shallow settings to deep settings. In the case of frequency, it goes from high to low with increasing depth; for ping mode, we see Frequency Modulated (FM) mode engaged toward the deeper range of the system; for depth mode, we see the system step from very shallow up to very deep mode. While these trends are generally expected, these particular plots also display a curious lack of uniformity in their progression, namely with regard to the switching of mode two shown in Figure 10c. When compared against the associated bathymetry, these extinction lines hinted at either a malfunctioning sonar or a misconfiguration of the operating parameters.

This prompted an examination of the sonar settings to determine if anything was amiss. We determined that the Angular Coverage mode was set to Manual, fixing the swath angle instead of allowing it to dynamically adjust based on the operating conditions and depth. This setting, as well as all other runtime settings, can be seen in the Kluster Attribution window, shown in Figure 11.

With the manual Angular Coverage mode limiting the performance of the sonar and resulting in poor outer beam performance, a second test was planned to determine the appropriate swath width relationship. This test is shown below in Figure 12. The mode change resulted in a much cleaner swath as the system compensated for depth by adjusting the beam angles appropriately. The system did not perform as well as expected, both in terms of ultimate depth range and swath width in deeper waters, but due to the heavy weather seen during this test, it was not entirely unexpected. Additionally, in the second extinction test, the minimum frequency was intentionally set to 70 kHz, instead of 50 khz, resulting in a narrower swath in deeper depths, as compared to the first test. This is reflected in Figures 10a and 12a.

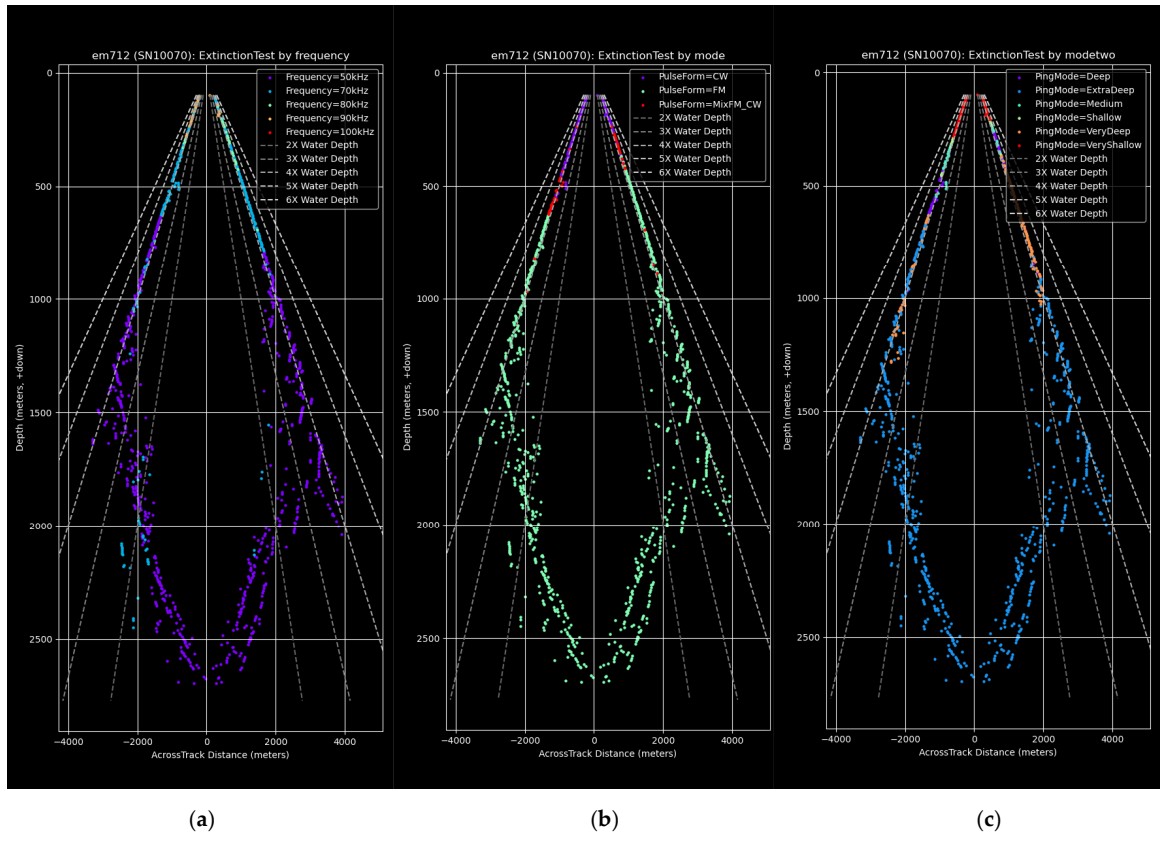

(**a**)  (**b**)  (**c**)

**Figure 10.** Kluster Extinction Test results for the initial test. Illustrates: (**a**) Swath width versus depth colored by frequency; (**b**) Swath width versus depth colored by the first mode value, which is either a Continuous Wave (CW) pulse or a Frequency Modulated (FM) pulse; (**c**) Swath width versus depth colored by the second mode value, which is the Kongsberg Ping Mode.

| | Attribute | Value |
|---|---|---|
| 57 | | tx: unit vector |
| 58 | | x: reference |
| 59 | | y: reference |
| 60 | | z: reference |
| 61 | | heading: reference point |
| 62 | | heave: transmitter |
| 63 | | pitch: reference point |
| 64 | | roll: reference point |
| 65 | runtimesettings_1654148145 | ('{"Max angle Port": "70.0", "Max angle Starboard": "70.0", "Max coverage '... |
| 66 | runtimesettings_1654158768 | ('{"Max angle Po |
| 67 | runtimesettings_1654158772 | ('{"Max angle Po |
| 68 | runtimesettings_1654158795 | ('{"Max angle Po |
| 69 | runtimesettings_1654158843 | ('{"Max angle Po |
| 70 | runtimesettings_1654158882 | ('{"Max angle Po |
| 71 | runtimesettings_1654158899 | ('{"Max angle Po |
| 72 | runtimesettings_1654159038 | ('{"Max angle Po |
| 73 | runtimesettings_1654159046 | ('{"Max angle Po |
| 74 | runtimesettings_1654159062 | ('{"Max angle Po |

('{"Max angle Port": "70.0", "Max angle Starboard": "70.0", "Max coverage 'Port": "3000.0", "Max coverage Starboard": "3000.0", "Angular coverage 'Mode": "Auto", "Beam spacing": "High density", "Forced depth": "--", "Min. 'depth": "2.0", "Max. depth": "4500.0", "Dual swath": "Dynamic", "Freq. 'range": "70-100kHz", "FM disable": "Off", "Water column data": "On", "Pitch 'stabilisation": "On", "Transmit angle Along": "0.0", "Max. Ping Freq. (Hz)": '"40.0", "External Trigger": "Off", "Yaw Stabilisation Mode": "Rel. mean 'heading", "Yaw Stabilisation Heading Filter": "Medium", "3D Scanning 'Enable": "Off", "Sound Velocity source": "Probe", "Sensor Offset": "0.0", '"Filter": "60.0", "Spike filter strength": "Medium", "Range gate size": '"Normal", "Phase ramp": "Normal", "Penetration Filter Strength": "Medium", '"Slope": "On", "Aeration": "Off", "Interference": "Off", "Normal incidence 'corr.": "10.0", "Use Lambert\'s law": "On", "Transmit power level": '"Normal", "Soft startup": "0", "Water Column": "On", "Water column": "30 dB 'Offset", "Water phase data": "Off", "Sonar mode": "Off", "Extra detection": '"Off", "Enable Simulation": "Off", "Scope display enabled": "Off", '"Counter": ""}')

**Figure 11.** Runtime parameters as shown in the Kluster Attribute window. Illustrates the runtime parameters for the second extinction test, where the Angular Coverage was changed to Auto.

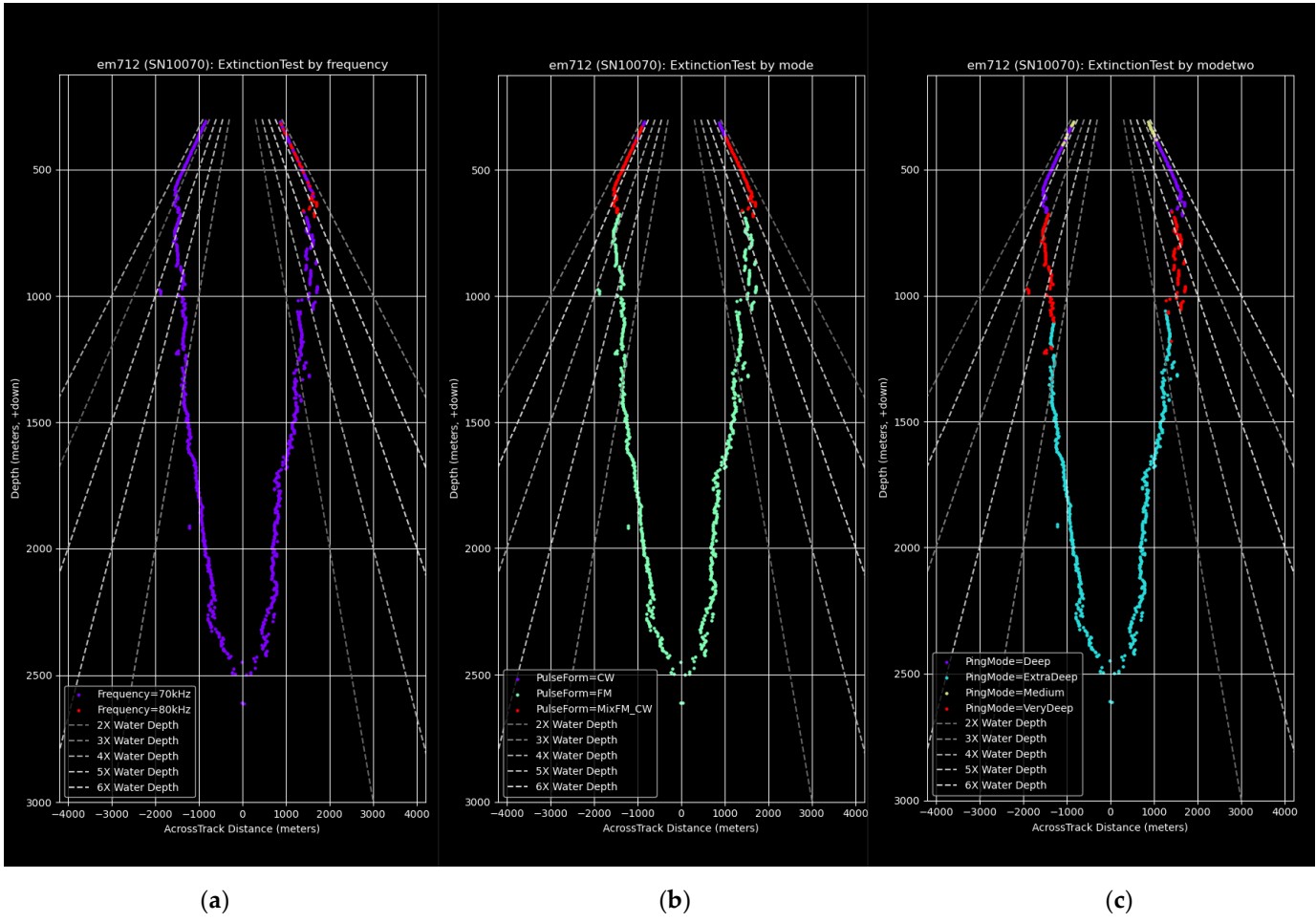

(**a**)          (**b**)          (**c**)

**Figure 12.** Kluster Extinction Test results for the second test. Illustrates: (**a**) Swath width versus depth colored by frequency; (**b**) Swath width versus depth colored by the first mode value, which is either a Continuous Wave (CW) pulse or a Frequency Modulated (FM) pulse; (**c**) Swath width versus depth colored by the second mode value, which is the Kongsberg Ping Mode.

### 3.4. Accuracy Test

The accuracy test is a means of determining the internal consistency of the sonar's seafloor measurements as a function of the beam angle, looking specifically for vertical differences between reference and acquired data. This is accomplished by comparing test lines against a densely populated, gridded dataset. First, we drove parallel lines to acquire a high density dataset over a relatively flat seafloor area. This dataset populated a reference bathymetry grid, against which we compared our accuracy lines. The accuracy lines were driven orthogonal to the set of reference tracklines, in pairs associated with each frequency and depth mode. In this way, the accuracy test allows the sonar operator to identify and isolate areas of high depth uncertainty and determine if they are a function of mode, frequency, or beam-angle. NOAA Ship *Fairweather* collected accuracy test data in multiple modes, frequencies, and depth regimes. In this paper, we focus on the accuracy test results gathered in roughly 250 m of water, in medium depth mode.

After processing the raw mutlibeam data, we imported post-processed navigation which automatically initiated a new cycle of georeferencing for the lines. Processing the accuracy test itself in Kluster occured in two independent steps. First, we selected the lines associated with the reference grid and created a surface. In this case, we created a variable resolution grid with depths computed by the CUBE algorithm. We then selected the accuracy test lines and used the Advanced Plots tool to conduct a grid-to-sounding comparison and output our beam-wise and angle-wise comparison plots. Typically, ac-

curacy test lines are run in succession as the sonar operator shifts through the various frequencies and modes. Kluster automatically groups these lines by frequency and mode before conducting the comparison. In this example, we selected a total of four lines, two in 70–100 kHz medium mode and two in 70–100 kHz deep mode. Kluster outputs accuracy plots corresponding to each pair respectively. The depth bias plot as a function of beam angle for 70–100 kHz in medium depth mode is shown below in Figure 13.

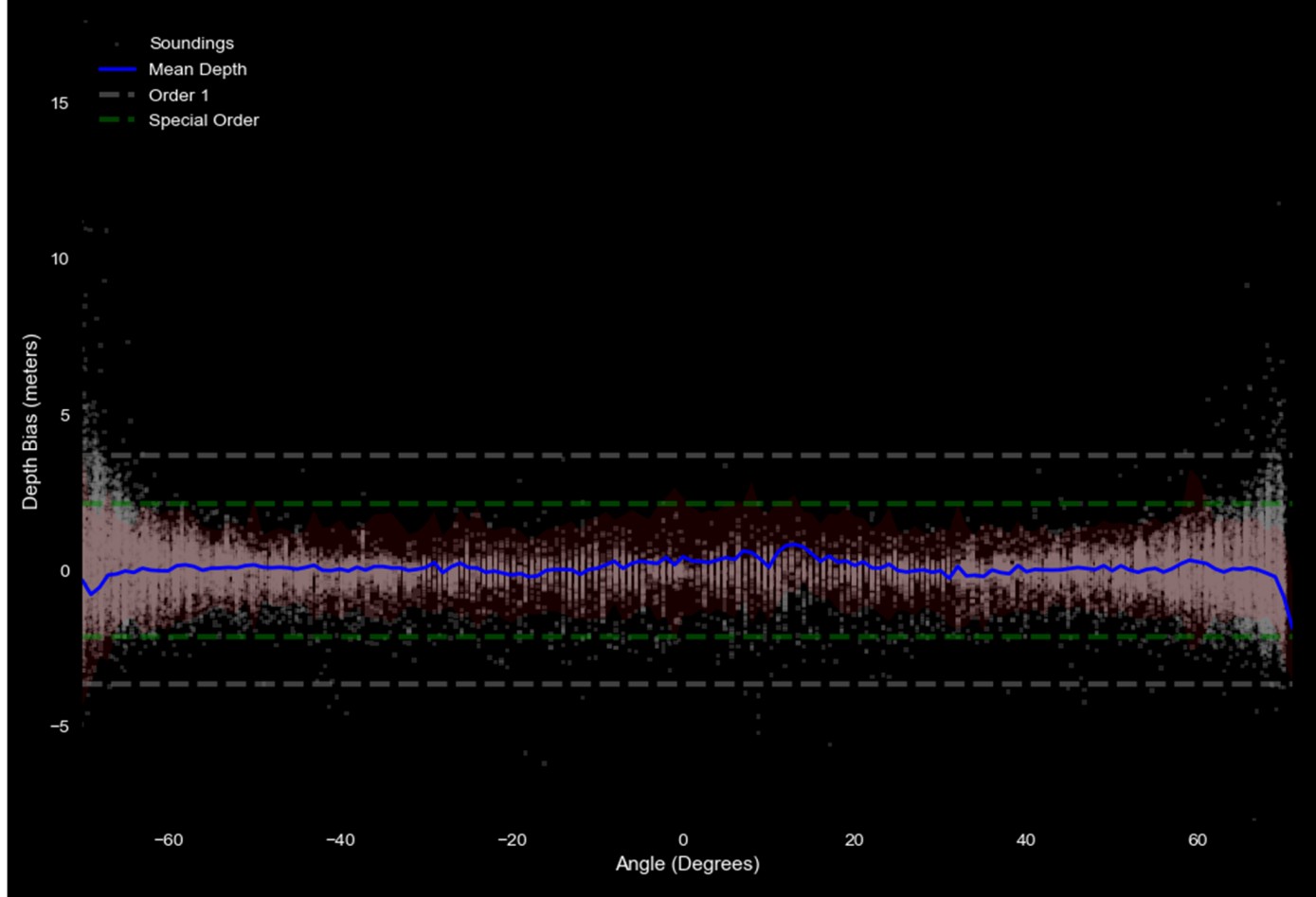

**Figure 13.** Kluster Accuracy Test result, showing the depth bias between the accuracy test lines in 70–100 kHz, medium mode, and the reference grid, plotted as function of beam angle. Comparisons to IHO Order 1 and IHO Special Order are shown as the horizontal dotted lines.

When plotting the grid-to-sounding comparison, Kluster automatically computes the average depth offset between the accuracy soundings and the reference surface. This bias is then removed from the computed result and displayed at the top of the chart, which enables a more coherent visualization of the small differences in depth bias as a function of angle or beam. In this example, the average bias was −3.5 cm, meaning the soundings were, on average, 3.5 cm shallower than the reference grid. The additional plotting of the Order 1 and Special Order specification for the reference surface depth allows us to quickly confirm that this sonar meets the requirement for uncertainty in the displayed mode.

## 4. Discussion

Kluster provides a new and intuitive way to process and analyze multibeam data. Through the integrated intelligence module, Kluster will spawn the appropriate actions for the user to take. This eliminates the need for the user to intimately understand the idiosyncrasies of multibeam processing, as is commonly required by other existing software. Experienced hydrographers and inexperienced users seeking to access multibeam data

alike stand to benefit from this simplified workflow. Through the Sonar Acceptance Test toolset, Kluster also supports analysis of the state of the sonar itself, in a way not currently available in other open-source software.

Kluster currently includes support for only Kongsberg (.all, .kmall, .raw) and Reson (.s7k) formats, but with the intermediate Xarray/Zarr format that Kluster generates on data conversion, the system has the capacity to support other sonar systems in the future. This future work item is a high priority of the project, as it directly enables the growth of the community around the software package. Kluster is available on a publicly accessible GitHub repository [6]. New releases include Windows builds of the software package for users unfamiliar with the creation of a Python environment.

Being entirely written in Python, Kluster is an attractive project for developers of all skill levels, as interacting and building off Kluster is a relatively simple matter. With the development of plugins, such as the Filter Module in Kluster, community engagement with Kluster can be made even easier—supporting experimentation in multibeam processing in a new and exciting way. Kluster represents a valuable tool for universities, companies, governments, or even individuals that seek to process multibeam data; be it for education, crowd sourced bathymetry, charting acquisition, or any other application of seafloor data.

**Author Contributions:** Conceptualization, E.Y.; methodology, E.Y.; software, E.Y.; validation, E.Y.; formal analysis, E.Y. and S.H.U.; investigation, E.Y. and S.H.U.; resources, E.Y.; data curation, E.Y. and S.H.U.; writing—original draft preparation, E.Y. and S.H.U.; writing—review and editing, E.Y. and S.H.U.; visualization, E.Y.; supervision, E.Y.; project administration, E.Y. All authors have read and agreed to the published version of the manuscript.

**Funding:** This research received no external funding.

**Institutional Review Board Statement:** Not applicable.

**Informed Consent Statement:** Not applicable.

**Data Availability Statement:** Access to Kluster is detailed in Section 2. The EM712 acceptance test dataset is unavailable for public download at this time.

**Acknowledgments:** We would like to extend our gratitude to the crew of NOAA Ship *Fairweather* for their dedication of time, resources, and energy to the acceptance testing of their EM712 sonar and the usage of Kluster.

**Conflicts of Interest:** The authors declare no conflict of interest.

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
