# Peer review of "Performing a Sonar Acceptance Test of the Kongsberg EM712 Using Open-Source Software: A Case Study of Kluster"

_2673-7418, doi:10.3390/geomatics2040029_

Round 1

Reviewer 1 Report

no

Author Response

No comments or suggestions provided.

Reviewer 2 Report

To the authors,

This paper has strong significance to the marine science community. It is important there is an effective, open source multibeam processing application that can enable organisations and individuals to get the best out of their data and not be prohibited by expensive licensing. It is also great to see that Kluster meets the requirements for calibrating precise survey equipment.

I see no issues with the science itself or experiment design, but the presentation of this paper could be greatly improved (please see attached document for more details) that when rectified, can turn it into a go-to resource:

1. Much of what is presented in the results is not results, but rather methods and discussion. Making the methods and discussion sections larger can give more confidence into the validation of Kluster. Results only need to include findings from the tests. Interpretations of the results go in the discussion.

2. Some of the figures are not necessary and other figures need tidying/editing. Clear presentation is essential and some of the figures have many windows and small text. Adding a map at the beginning will help situate where the study area is and give the reader spatial context.

3. There is no discussion/conclusion. The discussion (section 4) reads like a conclusion but it needs to be interpretation of the results. If the content in section 4 is made more concise, it would make a good conclusion. 

 There are other minor adjustments that can be made:

1. This paper includes very few references. Adding more can help with the robustness of the work. I have highlighted areas that could use additional references (attached).

2. Inconsistent use of first person. It can be simpler to write the document as third person and be wary of tense.

3. Minor grammatical changes to improve clarity and conciseness.   

This may not be the result you were looking for, but I hope you can make these changes because this is important work and this paper can be significant to the global marine science community. I would be happy to review again after the changes are made.

Author Response

This paper has strong significance to the marine science community. It is important there is an effective, open source multibeam processing application that can enable organisations and individuals to get the best out of their data and not be prohibited by expensive licensing. It is also great to see that Kluster meets the requirements for calibrating precise survey equipment.

I see no issues with the science itself or experiment design, but the presentation of this paper could be greatly improved (please see attached document for more details) that when rectified, can turn it into a go-to resource:

1. Much of what is presented in the results is not results, but rather methods and discussion. Making the methods and discussion sections larger can give more confidence into the validation of Kluster. Results only need to include findings from the tests. Interpretations of the results go in the discussion.

The reviewer made some very helpful wording suggestions, which I have corrected.  I do not currently have the time to restructure the paper as suggested, but I do understand the reviewer's perspective on this point.

2. Some of the figures are not necessary and other figures need tidying/editing. Clear presentation is essential and some of the figures have many windows and small text. Adding a map at the beginning will help situate where the study area is and give the reader spatial context.

Much of the data is inaccessible by me at this point, as I have moved to another position.  I would not be able to reload and make these figures.

3. There is no discussion/conclusion. The discussion (section 4) reads like a conclusion but it needs to be interpretation of the results. If the content in section 4 is made more concise, it would make a good conclusion. 

See my response to Item #1.

 There are other minor adjustments that can be made:

1. This paper includes very few references. Adding more can help with the robustness of the work. I have highlighted areas that could use additional references (attached).

I have added two references (those also suggested by another reviewer).

2. Inconsistent use of first person. It can be simpler to write the document as third person and be wary of tense.

I have made most of these corrections.

3. Minor grammatical changes to improve clarity and conciseness.   

I have made most of these corrections.

This may not be the result you were looking for, but I hope you can make these changes because this is important work and this paper can be significant to the global marine science community. I would be happy to review again after the changes are made.

My edited version is attached.

Thanks,

Eric

Reviewer 3 Report

Hello,

Thank you for writing this article. It gives a good understanding of the capabilities of the Kluster software and makes you want to test it. In the attached PDF you will find some proposals that improve, in my view, the article. The most important point according to me is the reference 6 which should be more adapted (with an author from Ifremer).

Have a nice day,

Author Response

Thank you for writing this article. It gives a good understanding of the capabilities of the Kluster software and makes you want to test it. In the attached PDF you will find some proposals that improve, in my view, the article. The most important point according to me is the reference 6 which should be more adapted (with an author from Ifremer).

Added hydrographic software keyword.

Added figure quotations

Add IHO S44 reference and CUBE reference

Added more information describing the titles of the sonar acceptance tests, and made recommended changes in the accuracy test section.

Made adjustments to References item 6

Round 2

Reviewer 2 Report

To the authors,

Thank you for considering the suggested grammatical changes.

I understand time pressures and other commitments can make thorough changes to manuscripts difficult to apply. However, I do not find the current state of this manuscript to be acceptable for publication. As suggested previously, appropriate content placing and clear figures are essential.

This manuscript may be better suited as government report if the changes logistically can not be made. The content is important and I would like to see it made available. 

All the best. 

Author Response

I have revised the images to make them easier to read.